# Exploring the Interplay between the Hologenome and Complex Traits in Bovine and Porcine Animals Using Genome-Wide Association Analysis

**DOI:** 10.3390/ijms25116234

**Published:** 2024-06-05

**Authors:** Qamar Raza Qadri, Xueshuang Lai, Wei Zhao, Zhenyang Zhang, Qingbo Zhao, Peipei Ma, Yuchun Pan, Qishan Wang

**Affiliations:** 1Department of Animal Science, School of Agriculture and Biology, Shanghai Jiao Tong University, Shanghai 200240, China; qamrq25@sjtu.edu.cn (Q.R.Q.); peipei.ma@sjtu.edu.cn (P.M.); 2Key Laboratory of Dairy Cow Genetic Improvement and Milk Quality Research of Zhejiang Province, College of Animal Science, Zhejiang University, Hangzhou 310030, China; laixsh3@sjtu.edu.cn (X.L.); 852322127@sjtu.edu.cn (W.Z.); 12117017@zju.edu.cn (Z.Z.); panyuchun1963@aliyun.com (Y.P.); 3Institute of Swine Science, Nanjing Agricultural University, Nanjing 210095, China; zhaoqingbo@njau.edu.cn; 4Hainan Institute, Zhejiang University, Yongyou Industry Park, Yazhou Bay Sci-Tech City, Sanya 572000, China

**Keywords:** hologenome, genome-wide association analysis, linear mixed models, complex traits, transcriptome-wide association studies, functional genomics

## Abstract

Genome-wide association studies (GWAS) significantly enhance our ability to identify trait-associated genomic variants by considering the host genome. Moreover, the hologenome refers to the host organism’s collective genetic material and its associated microbiome. In this study, we utilized the hologenome framework, called Hologenome-wide association studies (HWAS), to dissect the architecture of complex traits, including milk yield, methane emissions, rumen physiology in cattle, and gut microbial composition in pigs. We employed four statistical models: (1) GWAS, (2) Microbial GWAS (M-GWAS), (3) HWAS-CG (hologenome interaction estimated using COvariance between Random Effects Genome-based restricted maximum likelihood (CORE-GREML)), and (4) HWAS-H (hologenome interaction estimated using the Hadamard product method). We applied Bonferroni correction to interpret the significant associations in the complex traits. The GWAS and M-GWAS detected one and sixteen significant SNPs for milk yield traits, respectively, whereas the HWAS-CG and HWAS-H each identified eight SNPs. Moreover, HWAS-CG revealed four, and the remaining models identified three SNPs each for methane emissions traits. The GWAS and HWAS-CG detected one and three SNPs for rumen physiology traits, respectively. For the pigs’ gut microbial composition traits, the GWAS, M-GWAS, HWAS-CG, and HWAS-H identified 14, 16, 13, and 12 SNPs, respectively. We further explored these associations through SNP annotation and by analyzing biological processes and functional pathways. Additionally, we integrated our GWA results with expression quantitative trait locus (eQTL) data using transcriptome-wide association studies (TWAS) and summary-based Mendelian randomization (SMR) methods for a more comprehensive understanding of SNP-trait associations. Our study revealed hologenomic variability in agriculturally important traits, enhancing our understanding of host-microbiome interactions.

## 1. Introduction

Complex traits are crucial in livestock productivity, health, and adaptation. These traits are influenced by more than one factor due to the polygenic effect and complicated interplay of host genetics and environmental factors. Genome-wide association studies (GWAS) are the most widespread method used to study the sophisticated genetic architecture of complex animal traits [1]. They are a powerful tool that utilizes linear mixed models (LMMs) to identify single nucleotide polymorphisms (SNPs) or regions, assess statistical significance, and then fine-mapping specific regions to uncover functional variants associated with complex traits. LMMs are an extension of simple linear regression statistical models used to analyze data with fixed and random effects, allowing for more complex relationships between variables [2]. Recent studies have also extended the application of GWAS to microbiome-wide association studies (MWAS) and identified that the host’s microbiome plays an essential role in tailoring complex traits [3]. These studies leverage high-throughput metagenome sequencing to form a group or cluster of similar microbes based on their genetic similarities, called operational taxonomic units (OTUs). The OTUs are analyzed to associate microbial abundance and functional relevance with the host’s phenotype. In addition, OTUs also serve as an aggregation unit to reduce the dimensionality and sparsity of microbiome datasets [4]. However, GWAS and MWAS methods provide incomplete coverage of functional elements and miss the tissue-specific effects consisting of host-microbiome interaction, which are crucial for understanding complex traits for livestock breeding programs [5]. Furthermore, these methods also face challenges in identifying causative genetic effects and functional context due to their focus being solely on the host genome or microbial taxa [6,7].

Over the past few years, the hologenome concept has gained importance and undergone further exploration to understand the complex relationship between the host and microbiome. The hologenome refers to the combined genetic material of a host organism and its associated microbial community [8]. The study of host-microbiome interactions provides insights into their dynamic relationship and is called hologenomics. It allows the targeting of the host genome and microbiome, forming a functional unit, in oeder to better understand complex traits better. Several associations have already been identified in important animal traits considering the hologenome [9,10]. Similarly, the complex-trait prediction models consisting of hologenomics have been estimated to have a high prediction accuracy and reliability for cattle and pigs compared to exclusively using host genomic or microbiome data [11,12]. Therefore, integrating hologenomics and genome-wide association (GWA) analysis can provide valuable insights into the genetic basis of complex animal traits compared to traditional GWAS and MWAS methods. We refer to this novel integration method as Hologenome-Wide Association Studies (HWAS). 

In HWAS, we can successfully use the hologenome interaction effect to increase the power of identifying genetic variants by capturing the combined effects from both genetic sources. [13]. It can also improve the accuracy of phenotype prediction since incorporating hologenome interactions accounts for the effects of the microbiome that influence host gene expression, which in turn affects complex traits [14]. Most of these associations are likely to be expression quantitative trait loci (eQTL) in complex traits [15], and they bridge the gap between genetic variations and gene expression levels across tissues or cell types. Furthermore, eQTL can be analyzed by using methods such as fine mapping, colocalization analysis, transcriptome-wide association studies (TWAS), and summary-based Mendelian randomization (SMR) [16,17]. These methods are crucial in understanding the molecular mechanisms behind trait variability. Specifically, TWAS uses the statistical approach to combine eQTL information with GWA summary statistics to prioritize candidate genes associated with tissue-specific expression in complex traits. In contrast, using the Mendelian randomization principle, SMR provides causal inference by leveraging GWA summary statistics and eQTL data. The SMR model uses SNPs as an instrumental variable to test for the causative effect of gene expression on the phenotype. TWAS and SMR can explore hologenome-specific effects on gene expression and trait outcomes, thus providing valuable insights into the host-microbiome intricate interplay.

In this study, we performed four types of GWA analysis using genotype and microbiome datasets of real animals (cattle and pigs) to study the relationship between the host and complex traits. The four types of GWA methods were (a) the GWAS method, (b) the microbial GWAS (M-GWAS) method, (c) HWAS-CG (hologenome interaction estimated using the covariance between random effects genome-based restricted maximum likelihood (CORE-GREML) method), and (d) HWAS-H (hologenome interaction estimated using the Hadamard product method). We analyzed experimental variables of milk yield, methane emissions, rumen physiology traits from cattle, and gut microbial composition traits from pigs. For the remainder of this study, we will refer to different experimental variables as “traits”. Next, we annotated genes to the significant SNPs identified with each model to understand the biological mechanisms and functional pathways related to complex traits. Finally, a TWAS analysis was performed by integrating the GWA results with the eQTL data. We also validated our results using the SMR method to explore the influence of genetic variants on host gene expression. Our results provide new insights into the hologenomics underlying complex traits and elucidate host–microbiome genomic relationships in livestock.

## 2. Results

In total, fourteen complex traits related to milk production, methane emissions, rumen physiology in cattle, and gut microbiome composition in pigs were analyzed in this study. Four types of LMMs were used, including two simple models for testing only host genome and microbiome effects, i.e., GWAS and M-GWAS models, and two interaction models constituting hologenome effects to test the association between genetic variants and complex traits. The GWA analysis identified the different number of significant markers associated with complex traits when considering various statistical significance thresholds (Table 1). For example, if we consider the arbitrary threshold (*p* < 0.05) and Benjamini-Hochberg (FDR-BH) to declare a significant association, HWAS-CG identified the highest number of SNPs, i.e., 130,337 and 234 SNPs, respectively. Similarly, by adjusting the significant threshold according to the Bonferroni correction (BC) and Genome-wide significance threshold (GWST) methods, the M-GWAS identified 34 and 18 SNPs, respectively. Overall, the hologenome-based HWAS-CG identified the most trait-associated SNPs, especially under relaxed nominal thresholds, while the M-GWAS detected most trait-associated SNPs under stringent multiple testing correction.

### 2.1. Genome-Wide Association Analysis of Milk Yield, Methane Emissions and Rumen Physiology Traits in Cattle

The GWA analysis of milk, fat, protein, lactose, and fat–corrected milk (FCM) shows important SNPs correlated in each model with milk yield (Figure 1). The M-GWAS, HWAS-H, and HWAS-CG identified eight, five, and four significant SNPs associated with milk, respectively (Figure 1A). The M-GWAS identified the most significant SNP, UFL-rs134432442, at chromosome 14 (−log10 *p*-value = 8.78). None of the models could significantly associate any genomic variant with fat and protein (Figure 1B,C). While analyzing lactose, the M-GWAS identified the highest number of significantly associated SNPs, i.e., seven, whereas the HWAS-CG, HWAS-H, and GWAS identified four, three, and one SNPs, respectively (Figure 1D). The most significant SNP was the same as the one identified in the milk analysis with the M-GWAS, with a −log10 *p*-value of 8.58. In the case of FCM, only M-GWAS identified one SNP, BovineHD1700008152 (−log10 *p*-value = 6.35), nearest to the BC threshold value (Figure 1E). The details of the significant SNPs identified by each model are shown in Appendix A.

Four models were used to analyze the methane production (CH4 g/d), yield (CH4 DMI), and intensity (CH4 ECM) in cattle emissions (Figure 2). None of the models significantly associated any genomic variant with CH4 g/d and CH4 DMI (Figure 2A,B). While three SNPs were significantly associated with CH4 ECM according to the GWAS, M-GWAS, and HWAS-H, four SNPs were significantly associated with CH4 ECM by HWAS-CG (Figure 2C). The HWAS-H identified the most significant SNP, ARS-BFGL-NGS-3398 (−log10 *p*-value = 11.62) and ARS-BFGL-NGS-35483 (−log10 *p*-value = 11.62), on chromosome 16.

For the rumen physiology traits, the SNP Hapmap32352-BTA-153912 was significantly associated with acetate according to the HWAS-CG (−log10 *p*-value = 7.54) and GWAS (−log10 *p*-value = 7.19) on chromosome 17 (Figure 3A). Furthermore, the HWAS-CG successfully associated the SNP, ARS-BFGL-NGS-17066, located on chromosome 18, with propionate (−log10 *p*-value = 6.8) and wolin (−log10 *p*-value = 6.55) (Figure 3B,C). The GWAS, M-GWAS, and HWAS-H models failed to detect significant SNPs associated with most rumen physiology traits.

### 2.2. Genome-Wide Association Analysis of Gut Microbial Composition Traits in Pigs

The GWA analysis of gut microbial composition provided important insight into genomic variant associations with daily gain (DG), feed conversion (FC), and feed intake (FI) (Figure 4).

For DG, the HWAS-CG and HWAS-H models identified five and four SNPs, respectively. The most significant SNP identified with the M-GWAS was ASGA0023892 on chromosome 5, with a −log10 *p*-value of 7.97 (Figure 4A). For the FC trait, the GWAS and M-GWAS each detected seven significant SNPs, compared to just one SNP each for the HWAS models (Figure 4B). The top FC-associated SNPs from the GWAS included ASGA0069501, ASGA0101971, DRGA0009289, DRGA0009304, MARC0066941, MARC0073975, and MARC0075559, with a −log10 *p*-value of 7.12, at chromosome 9 and 15. For FI, the M-GWAS identified the most SNPs (eight significant SNPs), while the other models found seven each (Figure 4C). The same top SNPs from the GWAS for FC showed the strongest association for FI, with a −log10 *p*-value of 10.65. The details of significant associated SNPs identified with each model are shown in Appendix A.

### 2.3. Gene Ontology and Functional Analysis of SNPs in Complex Traits

The most significant genetic variants associated with complex traits were annotated with candidate genes using three criteria: entirely located within the gene region, partially overlapping with other genes, or adjacent (upstream and downstream) to regions within 100 kb. Table 2 shows a list of candidate genes along with their functions.

The most significant SNP, UFL-rs134432442, was identified by M-GWAS for the milk and lactose trait in cattle. This SNP was annotated to two genes: Cleavage and Polyadenylation Specific Factor 1 (*CPSF1*) and Solute Carrier Family 39 Member 4 (*SLC39A4*). The M-GWAS also identified the SNP, BovineHD1700008152, in FCM, which was annotated to the *7SK* gene, an RNA polymerase II transcription regulator. The HWAS-H model notably associated the SNP, ARS-BFGL-NGS-3398, with the CH4 ECM trait, and it was situated within the region of the Opticin (*OPTC*) gene. The acetate was significantly correlated with the SNP Hapmap32352-BTA-153912, which was annotated to the gene Glutamate Ionotropic Receptor AMPA Type Subunit 2 (*GRIA2*). Propionate and wolin were significantly related according to HWAS-CG with the SNP ARS-BFGL-NGS-17066, which was annotated to the gene F-Box Protein 15 (*FBXO15*).

For DG, the top significant SNP, ASGA0023892, was mapped to the Transcription Factor 20 (*TCF20*) gene with the M-GWAS. Moreover, the GWAS identified seven SNPs that are most robustly related to FC and FI. These SNPs were mapped to four protein-coding genes: Ectonucleotide Pyrophosphatase/Phosphodiesterase 6 (*ENPP6*), Contactin 5 (*CNTN5*), Rho GTPase Activating Protein 42 (*ARHGAP42*), and Interferon Regulatory Factor 2 (*IRF2*). The remaining three genes were long non-coding RNAs (lncRNA).

To understand their biological functions and implications, we performed a gene ontology-based functional analysis of the candidate genes annotated with the top significant SNPs from our GWA analysis of complex traits. The top biological terms and functional pathways that are highly enriching for complex traits are shown in Figure 5.

Regarding milk production traits (milk, lactose, and FCM), a total of 41 GO terms and 5 KEGG pathways were significantly enriched (FDR < 0.1). In particular, the most enriched cellular component term, GO:0032444 (Activin responsive factor complex), was associated with the Forkhead Box H1 (*FOXH1*) gene related to milk and lactose, and the most enriched KEGG pathway was related to fat digestion and absorption (Figure 5A). As for the methane emissions trait CH4 ECM, 11 GO terms and 10 Reactome pathways were enriched (FDR < 0.1). Specifically, the top molecular function term GO:0015459 (Potassium channel regulator activity) was linked to the Potassium Voltage-Gated Channel Interacting Protein 4 (*KCNIP4*) gene, and the most enriched Reactome pathway was neurofascin interactions (Figure 5B). Moreover, for rumen physiology traits, there were 35 GO terms and 15 KEGG pathways significantly enriched (FDR < 0.1). Most notably, the top biological process GO:0004971 (AMPA glutamate receptor activity) was associated with the *GRIA2* gene for acetate, and nicotine addiction was the most enriched KEGG pathway (Figure 5C).

Regarding gut microbial composition, 30 GO terms and 6 KEGG pathways were enriched for DG (FDR < 0.1). Most significantly, the top molecular function term GO:0016863 (Intramolecular oxidoreductase activity, transposing C=C bonds) was linked to the Dopachrome Tautomerase (*DCT*) gene and the most enriched KEGG pathway was tyrosine metabolism (Figure 5D). Furthermore, for FC and FI, 23 GO terms and 2 Reactome pathways were deemed significant (FDR < 0.1). In particular, the cellular component GO:1904694 (Neg. reg. of vascular associated smooth muscle contraction) was associated with the *CNTN5* gene, and the most enriched Reactome pathway was related to GPI-anchored proteins (Figure 5D).

### 2.4. Transcriptome-Wide Association Studies of Complex Traits

The key purpose of applying TWAS in our study was to combine eQTL information with GWA summary statistics to prioritize candidate genes associated with tissue-specific expression in complex traits. The unique set of genes identified with each model via the TWAS (P_TWAS_ ≤ 0.05) for each trait was shown in Appendix A. The top candidate genes associated with tissue-specific expression according to the TWAS analysis (P_FDR_ ≤ 0.05) are shown in Table 3 (Appendix A).

The TWAS analysis revealed significant associations across most traits using the HWAS-CG model. A total of 12 significant genes were detected for CH4 ECM, as well as two in acetate, five each in propionate and wolin, 19 for DG, and 29 for FC. Notably, the most significant genes linked to specific traits were Chromosome 9 C6orf163 Homolog (*C9H6orf163*) linked with the oviduct (P_FDR_ = 0.0027) for CH4 ECM; SPOC Domain-Containing 1 (*SPOCD1*) linked with the uterus for acetate (P_FDR_ = 0.022); Mediator Complex Subunit 29 (*MED29*) linked with macrophages for both propionate (P_FDR_ = 0.00021) and wolin (P_FDR_ = 0.00036); Interleukin 33 (*IL33*) linked with the colon for DG (P_FDR_ = 1 × 10^−6^); and APC Downregulated 1 (*APCDD1*) linked with the hypothalamus for FC (P_FDR_ = 0.0009). Furthermore, the M-GWAS identified 52 genes associated with both milk and lactose traits, and four genes linked to FCM. The most significant gene, Glycosylphosphatidylinositol Anchor Attachment 1 (*GPAA1*), was associated with muscle tissue for the milk (P_FDR_ = 1.1 × 10^−6^) and lactose (P_FDR_ = 7.7 × 10^−7^) traits, whereas Ras Association Domain Family Member 3 (*RASSF3*) was linked to macrophages for FCM (P_FDR_ = 0.0073). Additionally, HWAS-CG and HWAS-H detected 15 genes in fat and nine in protein, respectively. The most significant gene, Nuclear Receptor-Binding Protein 2 (*NRBP2*), was associated with blood for the fat trait (P_FDR_ = 0.00036), while Transcription Factor AP-2 Gamma (*TFAP2C*) was affiliated with embryos for the protein trait (P_FDR_ = 0.022). Next, the GWAS identified three and four significant genes for CH4 g/d and CH4 DMI, respectively. Particularly, Exostosin-Like Glycosyltransferase 1 (*EXTL1*) was the significant gene associated with CH4 g/d in the hypothalamus (P_FDR_ = 0.025), while Apolipoprotein B mRNA-Editing Enzyme Catalytic Subunit 3H (*APOBEC3H*) was identified for CH4 DMI in monocytes (P_FDR_ = 0.00178). Finally, HWAS-H identified six significant genes for FI, whereas the most significant gene, Sphingomyelin Phosphodiesterase 1 (*SMPD1*), was linked with hypothalamus (P_FDR_ = 0.016).

In addition, we investigated the GO enrichment of tissue-specific genes to highlight the complex trait etiology in livestock (Appendix A). Generally, the tissue-specific expression of genes identified in the milk yield traits was related to significant biological processes and cellular components related to cell component organization, immune response, and transcription activities (Appendix A). For the methane emissions traits, the tissue-specific genes were associated with significant terms of biological processes and molecular functions. These terms were linked to metabolic processes, cell signaling, membrane transportation and components, and cell-protein complexes (Appendix A). Similarly, the GO terms in the rumen physiology traits were associated with important biological processes and molecular functions, including cell development, cellular bindings, and brain development (Appendix A). Finally, the GO terms significantly enriched in the gut microbial composition traits covered biological processes and cellular components related to metabolic processes, intracellular components, and extracellular components (Appendix A).

### 2.5. Integrative Summary-Based Mendelian Randomization Analysis of Complex Traits

The SMR analysis presented comprehensive results highlighting the relationship between genetic variants and their impact on complex traits. We focused on the five most significant tissue-specific expressions identified in the TWAS analysis (certain traits demonstrated fewer associations). These findings were compared with the SMR results by integrating the GWA summary statistics with publicly available cis-eQTL data (Table 4 and Appendix A). All of the genes associated in the SMR analysis with cis-eQTL data were also present in TWAS results, but we are discussing the most significant associations identified by applying BC (α = 0.05, 0.05/number of tested probes) here.

Our SMR analysis identified four significant gene associations with tissues in milk on chromosome 14 (Figure 6), i.e., the expression of Diacylglycerol O-Acyltransferase 1 (*DGAT1*) associated with mammary tissue (−log10 P_SMR_ = 5.28) and the uterus (−log10 P_SMR_ = 5.23), *SLC39A4* (−log10 P_SMR_ = 3.74) and *ENSBTAG00000053637* (−log10 P_SMR_ = 3.66) associated with the ovary, and IQ Motif and Ankyrin Repeat-Containing 1 (*IQANK1*) associated with the muscle tissue (−log10 P_SMR_ = 3.58). Similarly, four significant gene associations with tissues were revealed for lactose on chromosome 14 (Figure 7). The expression of *DGAT1* was associated with the liver (−log10 P_SMR_ = 6.1) and the uterus (−log10 P_SMR_ = 5.17), and *SLC39A4* (−log10 P_SMR_ = 3.71) and *ENSBTAG00000053637* (−log10 P_SMR_ = 3.63) were associated with the ovary. For the FC trait, the SMR analysis significantly associated the expression of *ENSSSCG00000037808* and Membrane Frizzled-Related Protein (*MFRP*) with the frontal cortex (−log10 P_SMR_ = 3.71) and milk (−log10 P_SMR_ = 3.71), respectively (Figure 8).

Remarkably, *DGAT1* exhibited a significant association with the uterus in milk, passing the heterogeneity in dependent instruments (HEIDI) test (P_HEIDI_ ≥ 0.05). Additionally, in lactose, both *DGAT1* and *SLC39A4* showed associations with the uterus and ovary, respectively, and these associations also passed the HEIDI test (P_HEIDI_ ≥ 0.05). These findings suggest that *DGAT1* and *SLC39A4* are the most functionally relevant genes underlying GWA analysis hits.

## 3. Discussion

In this study, we performed GWA analysis using an LMM approach to explore the composite interaction networks between the host’s genome and microbiome in complex traits. Four types of LMMs, namely GWAS, M-GWAS, HWAS-CG, and HWAS-H, were considered for analyzing cattle’s milk yield, methane emissions, and rumen physiology traits. We also studied gut microbial composition traits in pigs to show how the host-microbiome relationship impacts complex traits in monogastric animals. Additionally, we integrated eQTL data with our GWA results using TWAS and SMR methods to prioritize candidate genes associated with tissue-specific expression in complex traits.

### 3.1. SNPs and Candidate Genes Identified for Milk Yield, Methane Emissions and Rumen Physiology Traits in Cattle

In our GWA analysis, many of the strong associations detected with each model support previously reported SNPs for the same or correlated traits. The M-GWAS was the most effective model for identifying the pleiotropic effect of seven SNPs located at the *Bos taurus* autosome (BTA) 14 in two dairy production traits, i.e., milk and lactose. The UFL-rs134432442 was the most significant SNP and was annotated to the *CPSF1* gene, which plays a key role in mRNA cleavage and polyadenylation during transcription [18]. Likewise, the SNP was also mapped to *SLC39A4*, an important zinc transporter gene [19]. Cochran et al. [20] have reported this SNP and *CPSF1* gene’s significant association with milk production traits such as milk and fat yield, and fat and protein percentage in Holstein cattle. In addition, consistent with our results, this study also identified milk production-associated SNPs annotated to cattle genes, including *DGAT1*, *FOXH1*, Rho GTPase Activating Protein 39 (*ARHGAP39*), Sphingomyelin Phosphodiesterase 5 (*SMPD5*) and Lymphocyte Antigen 6 Family Member K (*LY6K*). Pedrosa et al. [21] have reported variants in the genes *SLC39A4* and *7SK* significantly linked to milk yield traits in North American Holstein cattle. In our study, the SNP associated with the *7SK* gene was identified by M-GWAS and demonstrated a significant correlation with FCM. Similarly, the significant SNPs in the *LY6D* and *LOC100141215* gene regions correlated with milk and lactose by HWAS-H and HWAS-CG, respectively. These genes were previously linked to milk production and lactation curve parameters in Holstein dairy cows [22].

Among the traits evaluated for methane emissions, CH4 ECM was the only trait significantly associated with four SNPs. Manzanilla-Pech et al. [23] have associated 893 SNPs (*p* < 0.005) on BTA 16 with methane yield during a GWAS in dairy cattle. However, the primary aim of that study was to characterize which methane trait would be optimal for breeding goals. Notably, two significant SNPs we identified, annotated to the *OPTC* and Neurofascin (*NFASC*) genes by HWAS-H, also reside on BTA 16, thus providing some validation of our findings with the previous study. Furthermore, *KCNIP4* has been positively linked to traits related to milk production and adaptability via selection signature analysis in dairy cattle [24]. The gene link to adaptation can be used for sustainable agriculture practices, such as enhancing cattle resilience to changing climates and potentially reducing stress-related methane emissions.

For rumen physiology traits, we identified a significant SNP mapped to candidate gene *GRIA2* on BTA 17 for acetate by HWAS-CG exclusively. A prior GWAS in Angus cattle has identified a residual feed intake-associated SNP in *GRIA2*, corroborating our study’s association with the correlating complex trait [25]. For propionate and wolin, a strong linkage comes from the *FBXO15* gene, which harbored significant SNP solely identified with the HWAS-CG model. Although this gene has been associated with drip loss in pigs [26] and virulence of the human pathogen *Aspergillus fumigatus*, an environmental carbon recycler [27], no previous reporting of its association with rumen physiology exists.

### 3.2. SNPs and Candidate Genes Identified for Gut Microbial Composition Traits in Pigs

The GWA analysis of gut microbial composition traits in pigs identified the association of ten significant SNPs with DG and eight SNPs each for FC and FI. First, the most significant SNP in DG was identified with the M-GWAS and was mapped to *TCF20* on the *Sus scrofa* chromosome (SSC) 15. This gene has been reported in a genome-wide study to be sensitive in female mouse lungs under dietary α-tocopherol [28]. Second, HWAS-CG highlighted the four significant SNPs located on SSC 11 (mapped to G Protein-Coupled Receptor 180 (*GPR180*), *DCT* (two SNPs), and SMAD Family Member 9 (*SMAD9*)) and one SNP located on SSC 9 (mapped to *ENSSSCG00000014730*). In prior mammalian research, these genes have demonstrated key roles in modulating microbiome-disease relationships. For example, one study found that the *GPR180* gene exhibits altered expression when mice are fed high-fat diets, interacting with gut peptides to regulate hepatic lipid metabolism [29]. Additionally, the *DCT* gene has been linked with the gut microbiota through the melatonin pathway [30], and the expression of *SMAD9* has been linked to *Bacteroides dorei* in the context of Type 1 Diabetes [31]. The gene *ENSSSCG00000014730* has been previously reported as a candidate gene based on selection signature for meat quality in Jinhua pigs [32]. These findings in model organisms provide functional context to our results and suggest future directions for translating our genetic insights into a composite interaction of host genome and microbiome in livestock. Finally, HWAS-H prioritized four SNPs on SSC 4,13,1, and 12, annotated to DC-STAMP Domain Containing 1 (*DCST1*), *ENSSSCG00000041418* (lncRNA), Osteoclast Stimulating Factor 1 (*OSTF1*), and Ectonucleotide Pyrophosphatase/Phosphodiesterase 7 (*ENPP7*), respectively. Huang et al. [33] have suggested that *DCST1* is highly expressed in the hepatocellular carcinoma tissues and has been associated with chronic liver diseases. Lynegaard et al. [34] found that *OSTF1* was downregulated in the intestinal tissue samples of pigs fed a diet deficient in crude protein, thus suggesting a higher diversity and different composition of gut microbiota in pigs. Additionally, *ENPP7* has been shown to affect microbial and transcriptomic responses to irregular feeding in pigs, and it may modulate microbiome fluctuations caused by a variable nutrient intake [35].

We identified seven shared SNPs associated with both FC and FI with the GWAS model. Of these, four were mapped to coding genes *ENPP6*, *CNTN5*, *ARHGAP42*, and *IRF2*, while the remaining three were lncRNA and located in intergenic regions. From the protein-coding genes, *ENPP6* plays a role in muscle control and memory through Choline metabolism [36], *CNTN5* is involved in cell adhesion molecules during neural development [37], *ARHGAP42* regulates angiogenesis, and *IRF2* acts as regulatory gene during Luminal epithelium formation [26]. A previous study reported that *ENPP6* expression was reduced in Angus cows suffering from fescue toxicosis, affecting animal feed intake and weight gain [38]. *CNTN5* expression was also downregulated by lipopolysaccharide challenge in a study on porcine intestinal epithelial cell barrier function [39]. Additionally, SNP in *ARHGAP42* was linked to hypertension in a study on the interaction between genomics and diet in male adults [40]. Similarly, *IRF2*, which is involved in immune responses and inflammation, may influence the gut microbial communities in pigs by modulating the host immune system and the intestinal barrier function [41].

The candidate genes annotated to the significant SNPs identified by all GWA analyzes suggest that the host genetic factors may affect the gut microbial composition. Further studies are needed to elucidate the molecular mechanisms and causal relationships underlying this complex interplay.

### 3.3. Dissecting Complex Traits—TWAS and SMR Analysis of Tissue-Specific Gene Expression

We have combined our GWA results with publicly available cis-eQTL data via TWAS and SMR analysis to enhance the discovery of putative causal genes underlying complex trait variation and provide biological context for interpreting the results.

The candidate genes identified with each model via the TWAS (P_TWAS_ < 0.05) for complex traits were broadly consistent with the GWA and SMR results, with the addition of new candidate genes highly associated with complex traits (Table 3 and Appendix A). Several cattle studies have already associated genes and their specific tissue expression with milk yield [42]. For example, Buaban et al. [43] observed *LRRC14*, *GPAA1*, *NRBP2*, *DGAT1,* and *SLC39A4* to be highly associated with milk production in Thai dairy cattle. In our tissue-specific gene expression study, the TWAS analysis significantly associated *LRRC14* with macrophages for the milk and lactose traits, *GPAA1* with muscle tissue for the milk and lactose traits, and *NRBP2* with blood for the fat trait. Similarly, the expression of *DGATI* was significantly associated with leukocytes, uterus, and blood in milk, fat, and lactose. The gene *SLC39A4* was unavailable in the TWAS gene expression panel; therefore, the association was missing. However, our SMR analysis of milk and lactose traits significantly associated *SLC39A4* with the ovary. Notably, the *SLC39A4* also passed the HIEDI test, suggesting it to be an important functional relevant gene for livestock breeding strategies. More importantly, the TWAS analysis using the HWAS-H and M-GWAS models significantly associated *TFAP2C* with embryos in protein and *RASSF3* with macrophages in milk, lactose, and FCM, respectively. The expression of *TFAP2C* is related to milk production and mammary gland development in sheep, and is highly adaptable to climate resilience [44]. *RASSF3* has been identified as the top differently expressed gene in the bovine rumen microbiome [36]. These findings suggest that manipulating these tissue-specific gene expressions could be a viable strategy for managing milk production and reducing methane emissions in ruminant livestock.

For methane emissions traits, the TWAS analysis significantly revealed the association of *EXTL1* with hypothalamus and macrophage in CH4 g/d, *APOBEC3H* with monocytes in CH4 DMI, and *C9H6orf163* with oviduct in CH4 ECM. Cheng et al. [37] have reported that the differential expression of *EXTL1* is linked to the high concentration of grass-silage diet in dairy cattle. Similarly, *APOBEC3H* has been linked to lactation [45] and *C9H6orf163* with rumen microbiome in cattle [46]. Our SMR analysis of CH4 g/d also associated the expression of *EXTL1* with the hypothalamus, and *AOX1* was linked with adipose and hypothalamus. Interestingly, in a cattle transcriptome study of ruminal epithelia, the expression of *AOX1* has been correlated with dietary changes [47]. The findings indicate a potential pathway for mitigating methane emissions through dietary modulation.

The TWAS analysis of rumen physiology traits significantly linked *SPOCD1* with uterus in acetate and *MED29* with macrophage in propionate and wolin. Notably, the gene expression of *SLC20A2* was associated with blood in propionate and wolin by SMR analysis. Given prior connections of *SPOCD1* with feed efficiency [48], *MED29*, and *SLC20A2* with rumen function in cattle [46,49], further investigation of these genes in metabolic pathways may provide insights into microbial composition impacting cattle feed digestion and production.

Finally, the TWAS analysis of pig gut microbial composition traits revealed associations: *IL33* with the colon in DG, *APCDD1* with the hypothalamus in FC, and *SMPD1* with the hypothalamus in FI. Choudhury et al. [50] have reported the association of *IL33* in the colon with pig’s gut mucosal immune system. Also, *APCDD1* was identified as a candidate gene for the fatness trait in the porcine model by integrating eQTL and genotyping data [51]. Similarly, the gene expression of *SMPD1* was linked to the intestinal transcriptome and gut microbiota in pigs with the same daily feed intake [35]. Furthermore, our SMR analysis identified significant associations of *ENSSSCG00000037808* and *MFRP* with frontal cortex and milk, respectively, in FC. The *ENSSSCG00000037808* is associated with liver in a transcriptomic profiling and diet comparison study in pigs [52], whereas *MFRP* has been correlated to weight and body gain traits in the Iberian crossbred pigs’ study [53]. Our tissue-specific associations of genes with the gastrointestinal tract, fatness, and feed intake can be utilized for more efficient diet management strategies, potentially lowering methane emissions and sustainable pig farming.

### 3.4. Limitations of the Analysis

Our study proves that incorporating microbiome and hologenome interaction into the GWA framework for complex traits is useful. However, a few limitations to the analysis need to be addressed for a more comprehensive study. First, we have tried to minimize the possibility of false positive or negative associations in our GWA study by using two types of datasets. The cattle dataset’s sample size is larger than the pig data. Still, we believe a larger sample size will improve statistical power and enhance the ability to detect genuine associations [54]. Second, the cattle dataset may exhibit an unaccounted population structure due to the samples’ different geographic origins. Although the PC technique was included in the LMMs, other statistical methods like ADMIXTURE can help to filter specific genomic regions associated with population structure. Third, our datasets are from publicly available real data that is not intended for GWA analysis of complex traits. For example, inflation is higher in HWAS-CG and HWAS-H than in the GWAS model during analysis of gut microbial composition traits. Measurement errors or the quality of the genetic data can impact our GWA results. Additionally, the modeling of hologenome interaction can be further improved by developing effective statistical methods and machine-learning algorithms [55]. Lastly, rapid changes occur in the hologenome response to the environment, which may influence the host’s gene expression or biological pathways. Novel approaches such as gene-based and pathway-based testing can increase the power of hologenome-based GWA analysis because they increase the power of association and require less statistical testing burden [56].

## 4. Materials and Methods

### 4.1. Animal Genotype and Metagenome Data

We used previously published datasets from two species that included cattle and pigs. Each dataset was used separately to perform GWA analysis of complex traits. The overview of the study is shown in Figure 9.

The first dataset comprised 796 Holstein dairy cows from UK and Italian farms [57]. The overall dataset consisted of Holstein and Nordic red dairy cows, but we only selected the Holstein cow breed because of the extensive hologenome data and complex trait record. Moreover, Holstein cows significantly impact the dairy industry, making them an ideal choice for studying hologenomics in complex traits [58]. In total, 138,892 SNPs were genotyped using the Bovine GGP HD chip v2 (150K), and 132,214 were assigned coordinates using UMC bovine coordinates. All cows had a call rate higher than 0.90, and more than 99% of SNPs had a call rate higher than 0.99. SNPs with minor allele frequency (MAF) > 5% were retained for further analysis. The amplicon sequencing of bacterial and archaeal 16S rRNA genes, ciliate protozoal 18S rRNA genes, and fungal ITS1 genes was done by using the MiSeq technology from Illumina (Fasteris, SA, Geneva, Switzerland). We downloaded the FASTQ paired-end reads files from Short Reads Archive (SRA), accession number PRJNA517480. The individual paired-end reads were pooled into a single FASTQ file to pick an OTU in the QIIME 2 (quantitative insights into microbial ecology) platform [59]. In total, 734 OTUs were retained at a 20% abundance rate.

The second dataset consisted of 207 German Piétrain sows and was genotyped using an Illumina PorcineSNP60 BeadChip [60]. The German Piétrain sows are an ideal subject for hologenomics, considering their efficient feed conversion ratio and meat quality [61]. Moreover, the phenotypes studied are advantageous for studying the impact of diet and microbial interactions on fat metabolism and the animal’s overall health. After removing SNPs with a call rate of less than 95%, MAF < 5%, and other quality control measures, 51,970 SNPs were kept for further analysis. The bacterial V1-2 region of the 16S rRNA gene was sequenced on an Illumina MiSeq platform. After the paired-end sequences’ bioinformatic processing and quality filtering, 1870 OTUs were filtered at a relative abundance greater than 0.01%.

### 4.2. Animal Phenotype Data

We analyzed milk yield, methane emissions, and rumen physiology traits in cattle. The animal diet was standardized as total-mixed rations (TMRs) and partial-mixed rations (PMRs), including grass silage or hay, maize silage, and concentrates. The milk samples were collected using infrared spectroscopy named FOSS MilkoScan (FOSS, Hillerød, Denmark) and were analyzed for milk yield (kg/d), fat (kg/d), protein (kg/d), lactose (kg/d), and FCM (kg/d). The methane emissions were measured through ruminal gas variables, i.e., CH4 production (g/ day), yield (g/kg DMI), and intensity (g/kg ECM). The rumen fluid was collected using a ruminal probe consisting of a perforated brass cylinder, reinforced flexible pipe, suction pump, and collection vessel. Rumen physiology was measured by the concentration of acetate (mmol/mol), propionate (mmol/mol), and wolin (mmol/mol) in the rumen.

From the pig dataset, feed conversion (kg) (FC), feed intake (kg) (FI), and daily gain (kg) (DG) were analyzed to study the gut microbial composition in the animal. The DG was calculated as the difference between the final and initial body weight divided by the number of days on the test, FC was calculated as the ratio of FI to DG, and FI was estimated by individual feed intake recording for group-housed pigs.

### 4.3. Genome-Wide Association Analysis and Linear Mixed Models

GWA analysis was performed with the package gaston v1.5.9 [62], implemented in R v4.3.0 [63]. The four (generalized) LMMs were used to test the association between genetic variants and complex traits by modeling fixed SNP and random polygenic effects. The models evaluated ranged from simple to complex, including (i) GWAS, considering genomic relationship matrix (GRM) as a polygenic effect; (ii) M-GWAS, comprising microbial relationship matrix (MRM) as a polygenic effect; (iii) HWAS-CG; and (iv) HWAS-H. The HWAS-CG and HWAS-H consisted of a hologenome relationship matrix (HRM) as a polygenic effect, estimated by using CORE-GREML and the Hadamard product method, respectively [64,65] (Figure 10). The eigen decomposition of GRM, MRM, and HRM was included as a variance-covariance matrix in LMM to run the analysis computationally fast. Separate models were run for the cattle and pig datasets, including separate fixed effects.

The first model was named GWAS because it only tests the host genomic effect in a GWA analysis framework. The model is as follows:
(1)
y=Xβ+Zγ+g+e

where *y* is the vector of complex traits; 
β
 is the vector of covariables with fixed effects including intercept, dietary components with first five principle components (PC) from cattle dataset, whereas age, slaughter weight at the test station, and first PC from pig dataset; 
γ
 is the vector of genotypes containing SNPs to measure the effect on *y*; 
X
 and *Z* are the corresponding design matrix; 
g
 is the vector of polygenic effect accounting for the random animal genomic effect assuming *g ~ N (0, G*
$\sigma_{g}^{2}$
*),* where *G* is the *GRM* of an individual’s SNPs and 
$\sigma_{g}^{2}$
 being the additive genetic variance, and 
e
 is the residual term 
$e\;\sim \;N\left(0,\sigma_{e}^{2}I\right)$
 with residual variance 
$\sigma_{e}^{2}$
. The *GRM* was estimated by using the ‘*GRM* ()’ function from the gaston package as:
(2)
$$GRM=\frac{1}{q-1}\;X_{S}X_{S}^{\prime }$$

where 
$X_{S}$
 is a standardized 
n×q
 genotype matrix, and *q* is the number of SNPs.

The Wald’s test statistic was used to test the significance of SNP effects and compute the *p*-values in GWA analysis, defined as:
(3)
$$W=\frac{{\hat{\gamma }}^{2}}{var(\hat{\gamma })}$$

where 
$\hat{\gamma }$
 is the SNP effect size estimate, and under the null hypothesis 
$H_{0}:\;\gamma =0$
, the Wald statistic follows approximately the chi-squared distribution with 1 degree of freedom (
$\chi_{1}^{2}$
) [66]

The second model was called M-GWAS because the host microbiome effect was included as a polygenic effect in the LMM. The model is as follows:
(4)
y=Xβ+Zγ+m+e

where 
y, Xβ, Zγ, and e
 are the same as described in model 1, and *m* is the vector of polygenic effect accounting for the random animal microbiome effect assuming *m ~ N (0, M*
$\sigma_{m}^{2}$
*),* with M being the MRM constructed from OTUs and 
$\sigma_{m}^{2}$
 being the microbial variance. The MRM was estimated as follows:
(5)
$$\mathrm{MRM}=\frac{RR^{\prime }}{n}$$

where *R* is the matrix with log-transformed prokaryote’s relative abundance from all animals and *n* is the number of OTUs within animals [60].

The third model was constructed by modeling the hologenome interaction using CORE-GREML as a polygenic effect and random animal genome and microbiome effect in the LMM. The model was labeled as HWAS-CG and is as follows:
(6)
$$y=X\beta +Z\gamma +g+m+h_{c}+e$$

where 
y, Xβ, Zγ, g, m, and e
 are the same as described in models 1 and 4 and 
$h_{c}$
 is the HRM constructed as follows:
(7)
$$h_{c}=\sqrt{M}.{\sqrt{G}}^{\prime }+(\sqrt{M}.{\sqrt{G}}^{\prime }{)}^{\prime }$$

where 
$\sqrt{G}$
 and 
$\sqrt{M}$
 are the Cholesky factorization of the G and M matrix, respectively, and 
$\sigma_{h_{c}}$
 is the covariance between the genomic and microbiome effects. The Cholesky factorization of the G and M matrix makes it useful for efficient numerical solutions, thus allowing explicitly estimated covariance between G and M.

The last model was labeled as HWAS-H. It was like the HWAS-CG except for the HRM, which was modeled using the Hadamard product. The model is as follows:
(8)
$$y=X\beta +Z\gamma +g+m+h_{hp}+e$$

where 
y, Xβ, Zγ, g, m, and e
 are the same as described previously, and 
$h_{hp}$
 is the HRM constructed by performing Hadamard product between the animal genome and microbiome effect. The hologenome covariance matrix was estimated as follows:
(9)
$$h_{hp}={\left(G\circ M\right)}_{ij}∶={\left(G\right)}_{ij}\cdot {\left(M\right)}_{ij}$$

where G and M are the genomic and microbial relationship matrix. The symbol 
“∘”
 is defined as the Hadamard product for the two matrices, and the symbol 
“∶=”
 means that the right-hand side defines the left-hand side, and G and M are multiplied entry-wise. The 
$h_{hp}$
 is a square matrix and 
$\sigma_{h_{hp}}$
 is defined as the covariance between the genomic and microbiome effects. The entry-wise control of the Hadamard product allows it to operate on each entry of G and M rather than on the whole matrix, which helps preserve the overall covariance structure of the hologenome effect [67].

### 4.4. Genome-Wide Association Analysis Threshold and Functional Genomics Analysis

The inflation of SNP effects due to multiple testing of SNPs adjacent to each other is critical to differentiate true SNP effects from false negative or positive effects. Various statistical methods are available to associate genomic variants with complex traits. For example, arbitrary threshold (*p* ≤ 0.05), false discovery rate control using Benjamini-Hochberg (FDR-BH) [68], Bonferroni correction (BC) [69], and Genome-wide significance threshold (GWST) [70]. We have applied these statistical methods to classify the genetic variants correlated with complex traits by each model. The FDR-BH was calculated using the ‘p.adjust (method = ‘bh’)’ function from the R language [71]. It offers a trade-off between power and false discovery risk, proving more suitable for exploratory studies or when power is a higher priority [72]. In our study, we prioritized minimizing false positives due to the involvement of hologenomics complexity. Therefore, the BC (0.05/n, n = number of SNPs tested) was applied to filter significant SNPs, determined by a *p*-value of 3.78 × 10^−7^ for the cattle dataset and a *p*-value of 9.62 × 10^−7^ for the pig dataset. BC allows us to control the overall false positive rate, crucial to maintaining our findings’ integrity and identifying SNPs with stringent multiple-testing corrections. Also, BC is statistically more conservative than the other methods due to the independence among every tested genetic variant [73], thus making it more reliable for controlling the overall family-wise error rate and hologenomic effect. The Manhattan plot was used to visualize the SNPs across the chromosome, and the Quantile-Quantile plot (Q-Q plot) was used to visualize the distribution of *p*-values (cutoff *p*-value < 0.05) by using package rMVP v1.0.8 [74].

The genome assemblies of cattle (ARS-UCD 1.2, GCA_002263795.2) and pig (Sus scrofa 11.1, GCA_000003025.6) were accessed from Ensemble release 110 to annotate significant SNPs [75]. Generally, we annotated SNPs with candidate genes using three criteria: entirely located within the gene region, partially overlapping with other genes, or adjacent (upstream and downstream) to regions within 100 kb. [76]. Next, we performed a gene ontology-based functional analysis to identify the enriched biological processes and pathways among our GWA hits. The ShinyGO 0.77 (FDR < 0.1) was used to obtain Gene Ontology (GO) terms, Kyoto Encyclopedia of Genes and Genomes (KEGG) pathways, and Reactome pathways [77], whereas the results were visualized by using SRplot, an online tool (http://www.bioinformatics.com.cn/srplot, accessed on 1 July 2023). When no KEGG pathways were available due to fewer genes, the Reactome pathways were analyzed.

### 4.5. Transcriptome-Wide Association Analysis

To enhance the understanding of hologenome interactions in complex traits, we used GWA summary statistics to perform TWAS analysis and gene set enrichment analysis (GSEA) by using the FarmGTEx TWAS-server (http://twas.farmgtex.org/, accessed on 5 July 2023) [78]. Firstly, the GWAS imputation module imputed the SNPs from summary statistics according to eQTL mapping of cattle and pig reference genomes. Secondly, the TWAS analysis was performed using the computationally efficient MetaXcan (S-PrediXcan) method [79]. A total of 34 and 23 tissues in pigs and cattle were scanned to perceive significant trait-underlying genes. The FDR < 0.05 was applied to highlight significant associations and adjust the multiple testing burden. Finally, GSEA was performed to investigate the important GO terms and interpret the biological mechanisms linked with complex traits. Venn diagrams were generated using the Venn web tool (https://bioinformatics.psb.ugent.be/webtools/Venn/, accessed on 10 July 2023) to visualize overlapping sets of significant genes (P_TWAS_ ≤ 0.05) prioritized by the TWAS analysis across each GWA model.

### 4.6. Summary-Based Mendelian Randomization Analysis

We conducted the SMR analysis using the SMR software version 1.3.1 to validate our GWA findings [17]. The SMR utilizes the Mendelian randomization principle to integrate GWA and eQTL summary statistics to test for the causative effect of gene expression on the complex trait. We primarily focused on the gene-tissue pair identified by our TWAS analysis for further validation (FDR ≤ 0.05). In order to test the association, SNPs from GWA analysis were treated as instrumental variables (Z), gene expression as exposure (X), and complex trait as outcome (Y). Therefore, the estimation of the SNP-trait effect using GWA summary data (b_ZY_) and the SNP-gene expression effect using eQTL summary data (b_ZX_) can be used for testing the association with complex traits (b_XY_ = b_ZY_/b_ZX_). We used the cis-eQTL summary statistics data from the pilot phase of the Cattle Genotype-Tissue Expression atlas (CattleGTEx) and Pig Genotype-Tissue Expression atlas (PigGTEx) (accessed on 2 April 2024) [42,80]. The cis-eQTL data from CattleGTEx consisted of 23 distinct tissues, processed from 7180 publicly available RNA-sequencing (RNA-seq) samples. Similarly, cis-eQTL data from PigGTEx consisted of 34 tissues and was processed from 5457 RNA-seq samples. The cis-eQTL summary files were converted to BESD format for reading in SMR software using the parameter ‘--fastqtl-nominal-format --make-besd’. We performed QC processing by applying a *p*-value of 1 × 10^−5^ (with a default window size of 2000 Kb) to select the top associated cis-eQTL and remove SNPs with discrepant allele frequencies between data sets (--diff-freq 0.9 --diff-freq-prop 0.1). The Bonferroni correction (0.05/n, n = number of probes tested) was applied to control the genome-wide type I error rate and account for multiple testing burdens. We further used a post-hoc method called the heterogeneity in dependent instruments (HEIDI) to test whether the association observed in SMR analysis is due to linkage or pleiotropy [17]. The SMR results passing the HEIDI threshold (P_HEIDI_ 
≥
 0.05) show evidence of no pleiotropic effects.

## 5. Conclusions

In this study, we provide valuable insights into the genetic basis of complex traits by integrating GWA analysis and hologenomics framework, i.e., HWAS. We used four types of LMMs, namely GWAS, M-GWAS, HWAS-CG, and HWAS-H, to study milk yield, methane emissions, rumen physiology in cattle, and gut microbiome composition traits in pigs. The M-GWAS was the most effective model to identify the pleiotropic effect of seven SNPs located at BTA 14 in two dairy production traits, i.e., milk and lactose. Notably, three significant SNPs identified by HWAS-H were significantly associated with CH4 ECM. Furthermore, HWAS-CG was the only model identifying significant SNPs in all rumen physiology traits. The genes annotated to these SNPs provide valuable information for sustainable agriculture practices, such as enhancing cattle resilience to changing climates, reducing stress-related methane emissions, and improving rumen function. In contrast, GWAS associated the highest number of genomic variants with gut microbial composition traits in pigs, and these associations were successfully mapped to genes actively involved in the composite interaction of the host genome and microbiome.

Finally, we performed TWAS and SMR analysis by integrating publicly available cattle and pig’s cis-eQTL data with our GWA results. Specifically, the association of *GPAA1*, *DGAT1*, and *SLC39A4* with milk and lactose, *EXTL1* with CH4 g/d, *MED29,* and *SLC20A2* with propionate and wolin, and *IL33* with DG highlighted vital findings related to milk production, methane metabolism, rumen function, and gut microbiota. These findings provide a viable strategy for managing milk production and reducing methane emissions using dietary modulation in cattle and pigs.

In conclusion, the HWAS-based GWA models provide a unique approach to integrating the host’s genome and microbiome for studying complex traits in cattle and pigs. However, the modeling of hologenome interaction can be further improved by developing effective statistical methods and machine-learning algorithms. Follow-up studies are warranted to validate genetic mechanisms and pathways connecting the hologenome to complex traits in agriculturally relevant species.

## Figures and Tables

**Figure 1 ijms-25-06234-f001:**
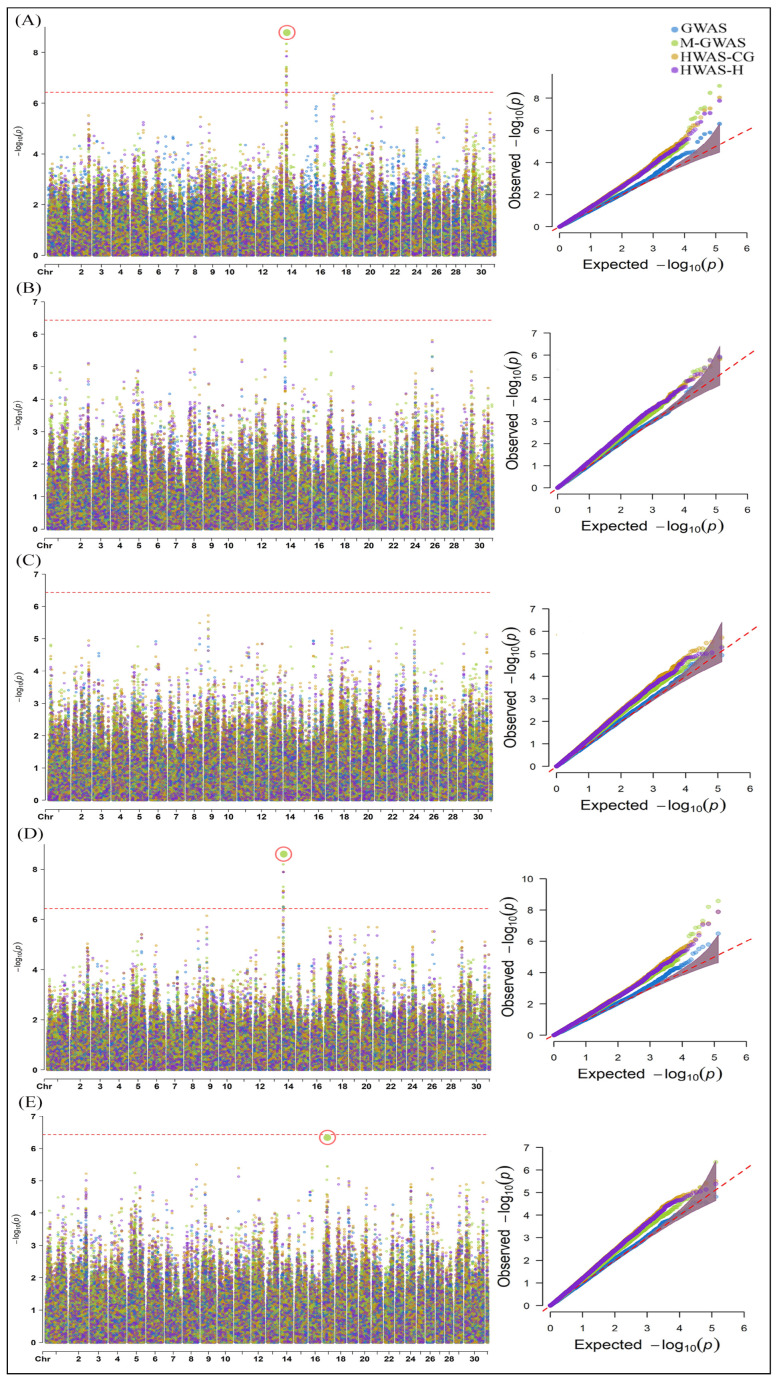
Manhattan and Q-Q plots of associated SNPs identified using GWAS, M-GWAS, HWAS-CG, and HWAS-H models for milk yield, including (**A**) milk, (**B**) fat, (**C**) protein, (**D**) lactose, and (**E**) FCM. The red-dotted line in the Manhattan plot represents the BC threshold (−log10 *p*-value = 6.42) and cutoff *p*-value < 0.05 in the Q-Q plot. The red circle in the Manhattan plot shows the most significant associated SNP according to the models.

**Figure 2 ijms-25-06234-f002:**
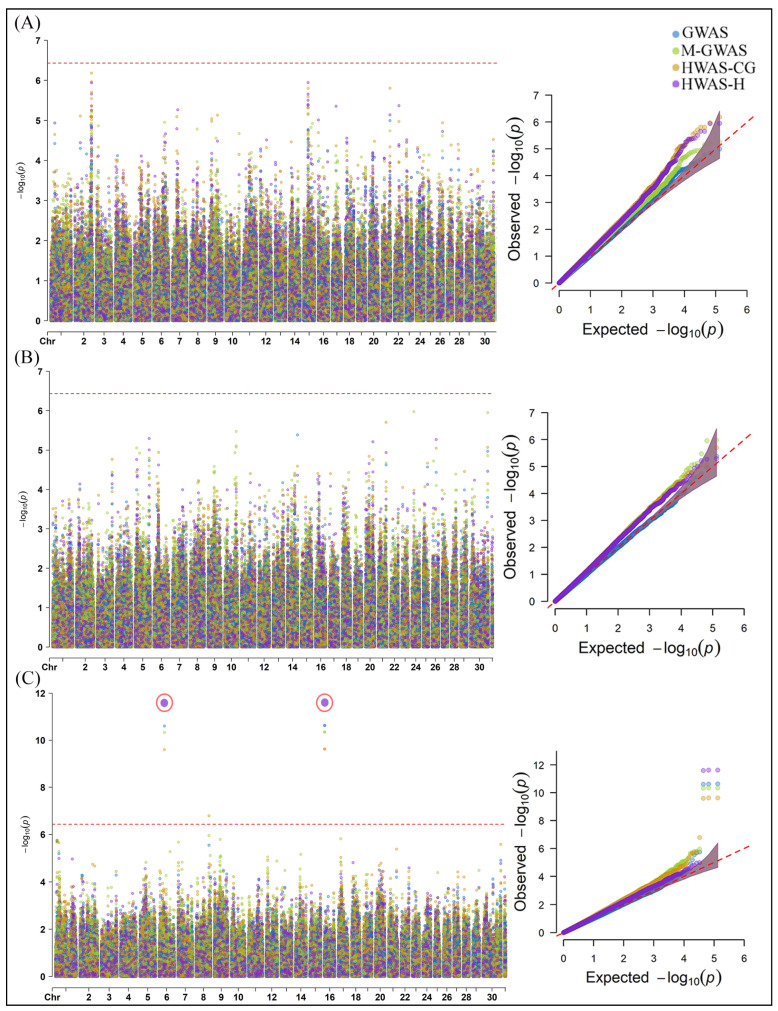
Manhattan and Q-Q plots of associated SNPs identified using GWAS, M-GWAS, HWAS-CG, and HWAS-H models for methane emissions, including (**A**) CH4 g/d, (**B**) CH4 DMI, and (**C**) CH4 ECM. The red-dotted line in the Manhattan plot represents the BC threshold (−log10 *p*-value = 6.42) and cutoff *p*-value < 0.05 in the Q-Q plot. The red circle in the Manhattan plot shows the most significant associated SNP according to the models.

**Figure 3 ijms-25-06234-f003:**
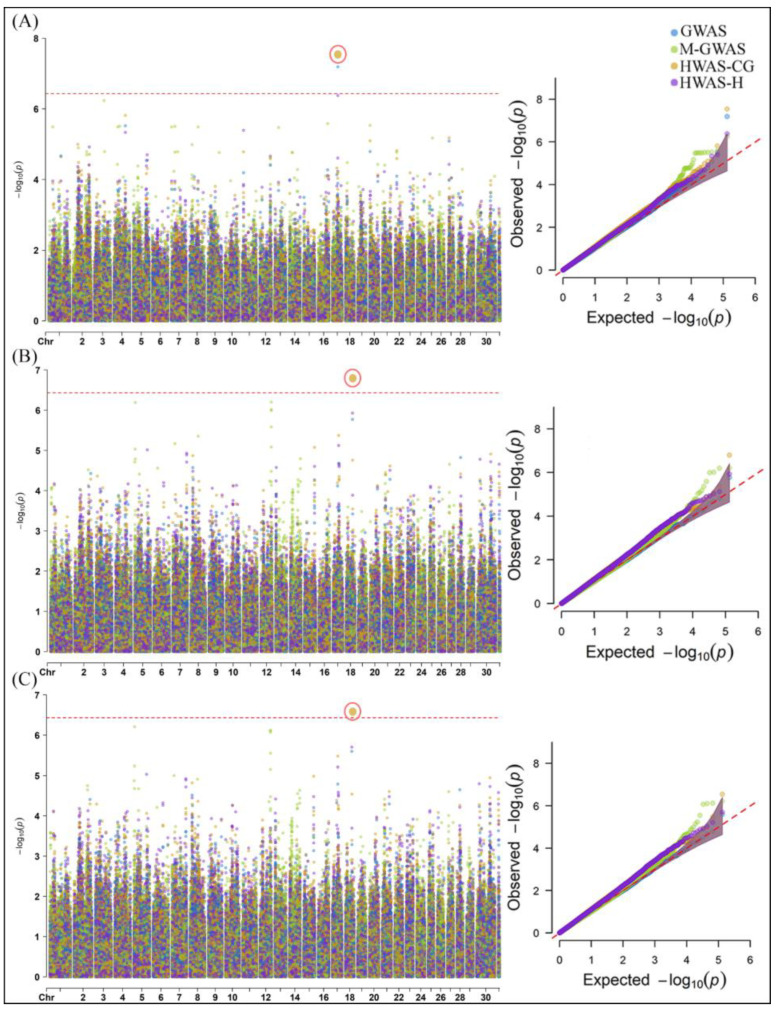
Manhattan and Q-Q plots of associated SNPs identified using GWAS, M-GWAS, HWAS-CG, and HWAS-H models for rumen physiology, including (**A**) acetate, (**B**) propionate, and (**C**) wolin. The red-dotted line in the Manhattan plot represents the BC threshold (−log10 *p*-value = 6.42) and cutoff *p*-value < 0.05 in the Q-Q plot. The red circle in the Manhattan plot shows the most significant associated SNP according to the models.

**Figure 4 ijms-25-06234-f004:**
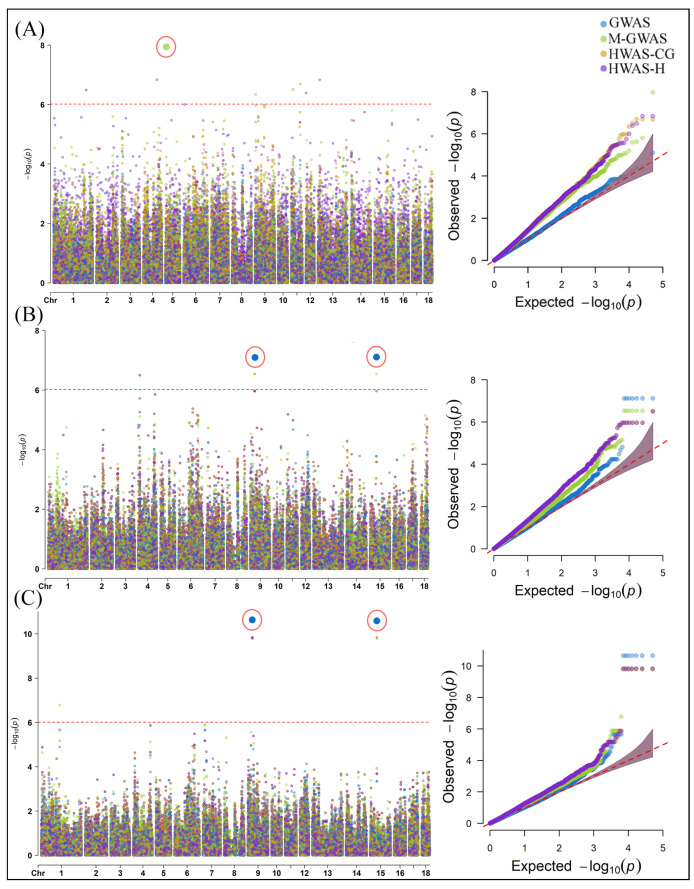
Manhattan and Q-Q plots of associated SNPs identified using GWAS, M-GWAS, HWAS-CG, and HWAS-H models for gut microbial composition, including (**A**) DG, (**B**) FC, and (**C**) FI. The red-dotted line in the Manhattan plot represents the BC threshold (−log10 *p*-value = 6.01) and cutoff *p*-value < 0.05 in the Q-Q plot. The red circle in the Manhattan plot shows the most significant associated SNP according to the models.

**Figure 5 ijms-25-06234-f005:**
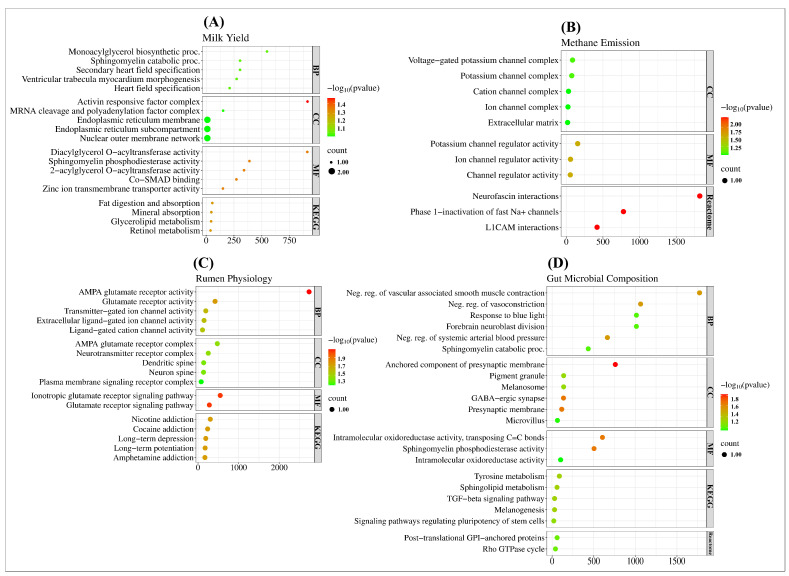
Gene ontology terms related to the biological process (BP), cellular component (CC), molecular function (MF), and functional pathways (KEGG and Reactome) of candidate genes in complex traits. The Bubble plot of the GO terms and functional pathways enriched in (**A**) milk yield, (**B**) methane emissions, (**C**) rumen physiology, and (**D**) gut microbial composition traits. The bubble color represents the FDR enrichment in −log10 scale, and the x-axis shows the fold enrichment score of the GO terms and functional pathways.

**Figure 6 ijms-25-06234-f006:**
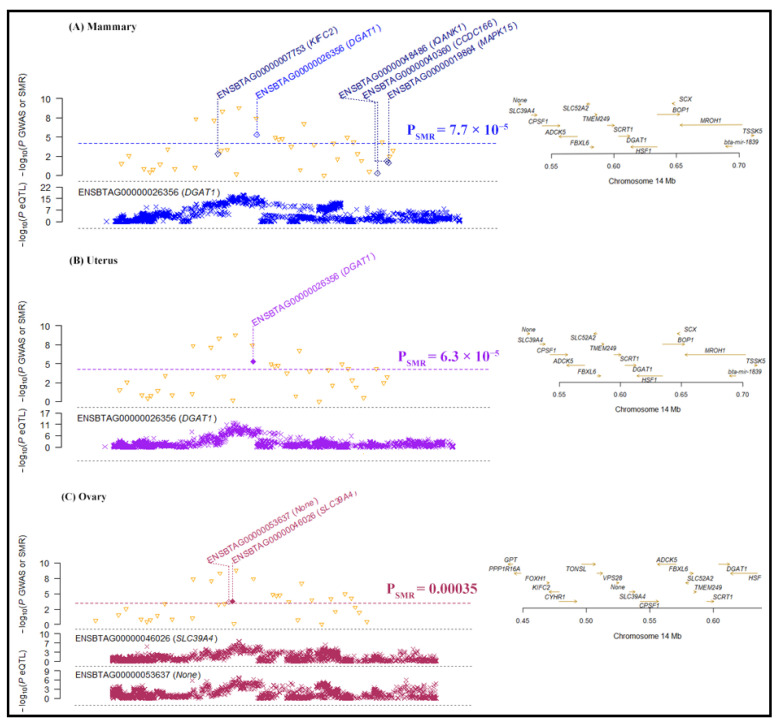
The genes associated with tissues in milk according to SMR analysis. The locus plot shows genes (yellow triangle) associated with (**A**) mammary, (**B**) uterus, and (**C**) ovary. The dotted line is the BC threshold, and solid rhombuses indicate the passing of the heterogeneity in dependent instruments (HEIDI) test. The middle plot shows the eQTL results, and the right plot shows the location of genes on the chromosome tagged using the probe.

**Figure 7 ijms-25-06234-f007:**
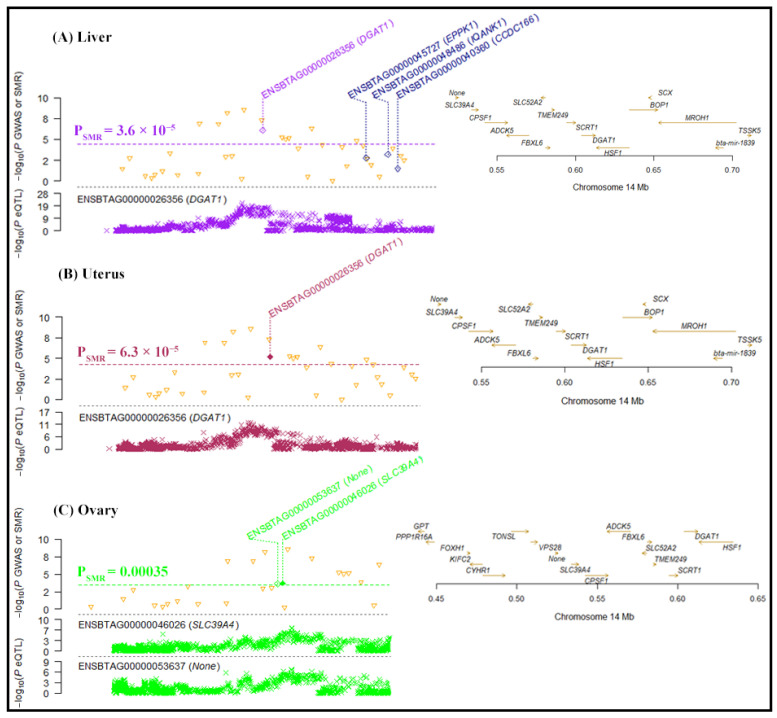
The genes associated with tissues for the lactose trait according to SMR analysis. The locus plot shows genes (yellow triangle) associated with (**A**) liver, (**B**) uterus, and (**C**) ovary. The dotted line is the BC threshold, and solid rhombuses indicate the passing of the HEIDI test. The middle plot shows the eQTL results, and the right plot shows the location of genes on the chromosome tagged using the probe.

**Figure 8 ijms-25-06234-f008:**
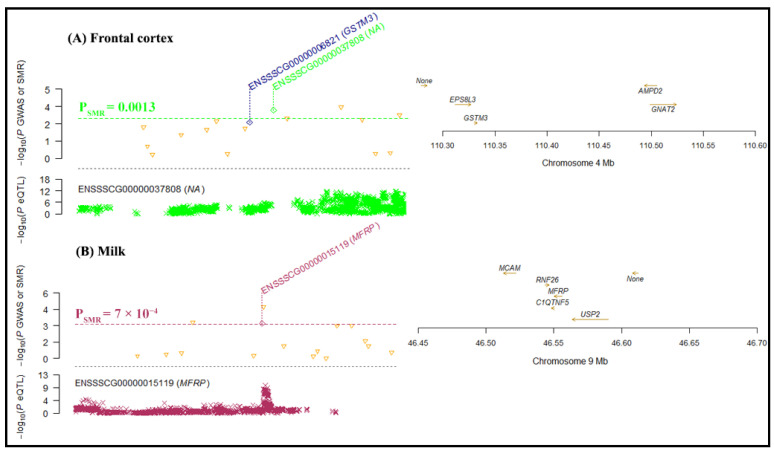
The genes associated with tissues for the FC trait according to SMR analysis. The locus plot shows genes (yellow triangle) associated with (**A**) frontal cortex and (**B**) milk. The dotted line is the BC threshold, and hollow rhombuses indicate the failure of the HEIDI test. The middle plot shows the eQTL results, and the right plot shows the location of genes on the chromosome tagged using the probe.

**Figure 9 ijms-25-06234-f009:**
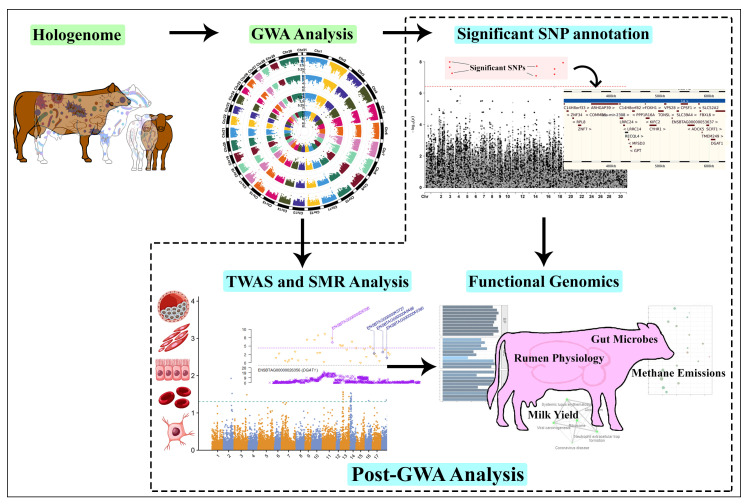
Visual representation of the study. The host hologenome was used to perform GWA analysis using separate linear mixed models. The significant SNPs identified by each model were mapped to candidate genes and further analyzed to highlight important GWA sites. TWAS and SMR analysis was performed using eQTL data to explore the tissue-specific expression of complex traits in livestock.

**Figure 10 ijms-25-06234-f010:**
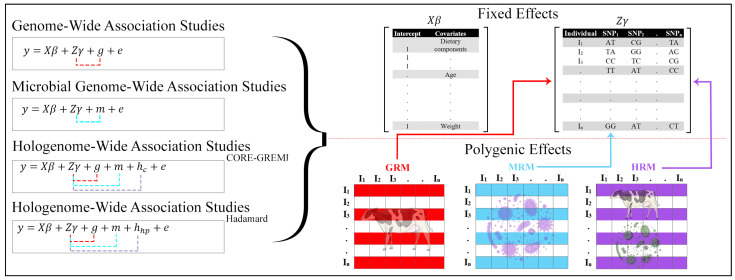
Overview of (generalized) LMMs used to test the association between genetic variants and complex traits. The left panel shows the full models used to test the association, i.e., genome-wide association studies using genomic effect (red dotted line), microbial genome-wide association studies using microbe effect (blue dotted line), hologenome-wide association studies (CORE-GREML) (purple dotted line), and hologenome-wide association studies (Hadamard product) (purple dotted line) using the interaction between genomic and microbe effects. The right panel shows the fixed effect matrices of covariates (including y-intercept), genotype, and relative interaction matrix (genomic relationship matrix (GRM), microbial relationship matrix (MRM), and hologenome relationship matrix (HRM)) used in each model as a polygenic effect to perform the GWA analysis of complex traits.

**Table 1 ijms-25-06234-t001:** Comparison of the number of significant SNPs identified in each GWA analysis according to various methods of determining statistical significance.

Model	Arbitrary Threshold	FDR-BH	BC	GWST
HWAS-CG	130,337	234	28	14
HWAS-H	124,212	118	20	12
M-GWAS	111,222	97	34	18
GWAS	84,281	29	19	10

**Table 2 ijms-25-06234-t002:** List of significant SNPs, annotated genes, and functions for the milk, lactose, FCM, CH4 ECM, acetate, propionate, wolin, DG, FC, and FI traits.

Trait	SNP ID	Gene	Function
Milk and Lactose	UFL-rs134432442	*CPSF1, SLC39A4*	mRNA cleavage and polyadenylation specificity factor complex, cellular zinc homeostasis as a zinc transporter *
	BovineHD1400000206	*SMPD5*	lipid and sphingolipid metabolism
	ARS-BFGL-NGS-4939	*DGAT1*	Absorption of dietary fats and esterification of exogenous fatty acids to glycerol
	BovineHD1400000188	*ARHGAP39*	Post-synapse organization and signal transduction
	ARS-BFGL-NGS-57820	*FOXH1*	Development of the mammary gland and the regulation of milk protein
	BovineHD1400000447	*LY6K*	Sperm migration and cell growth
	ARS-BFGL-NGS-103064		
	BovineHD1400000262	*LOC100141215*	Unknown function
Milk	BovineHD1400000453	*LY6D*	Marking the earliest stage of B- and T-cell development *
FCM	BovineHD1700008152	*7SK*	Negatively regulating RNA polymerase II transcription *
CH4 ECM	ARS-BFGL-NGS-3398	*OPTC*	Inhibiting angiogenesis and regulating collagen fibril organization *
	ARS-BFGL-NGS-35483	*NFASC*	Axon subcellular targeting and synapse formation *
	BTA-75940-no-rs	*KCNIP4*	Regulation of potassium ion transmembrane transport *
	ARS-BFGL-NGS-34028	*PALM2 ***	Membrane–cytoskeleton interaction
Acetate	Hapmap32352-BTA-153912	*GRIA2*	Various central nervous system functions *
Propionate and Wolin	ARS-BFGL-NGS-17066	*FBXO15*	Ubiquitin-mediated protein degradation *
DG	ASGA0023892	*TCF20*	Transcriptional regulator *
	ASGA0051282	*GPR180*	Thermogenic adipocyte activity *
	INRA0037215	*DCT*	Plays a role in melanin biosynthesis pathway *
	SIRI0001279		
	DIAS0003214	*SMAD9*	Cell signaling and act as transcription factors *
	M1GA0025187	*ENSSSCG00000014730*	Putative odorant or sperm cell receptor
	M1GA0006139	*DCST1*	Egg-sperm fusion and antigen processing
	ASGA0056172	*ENSSSCG00000041418*	None (Long non-coding RNA)
	DRGA0002098	*OSTF1*	Mastitis resistance and bone resorption activity
	MARC0008607	*ENPP7*	Protecting intestinal mucosa from inflammation
FC and FI	ASGA0069501	*ENPP6*	Synthesis of phosphatidylcholine and Choline metabolism *
	ASGA0101971	*ENSSSCG00000053826*	None (Long non-coding RNA)
	DRGA0009289	*CNTN5*	Cell-cell adhesion and brain development *
	DRGA0009304	*ARHGAP42*	Regulating endothelial cell shape and angiogenesis *
	MARC0075559		
	MARC0066941	*ENSSSCG00000053826*	None (Long non-coding RNA)
	MARC0073975	*IRF2*	Controlling the luminal epithelium of the endometrium
FC	MARC0057350	*ENSSSCG00000047619*	None (Long non-coding RNA)
FI	MARC0087957	*ENSSSCG00000045481*	Unknown function

* Studied in other mammals, ** Located within the 115kb region.

**Table 3 ijms-25-06234-t003:** A list of the top candidate genes associated in the TWAS with milk yield, methane emissions, rumen physiology in cattle, and gut microbial composition in pigs with their tissue-specific expressions.

Trait	Model	Gene	Chr	Start	End	P_FDR_	Tissue
Milk	M-GWAS	*GPAA1*	14	750,608	753,850	1.1 × 10^−6^	Muscle
Fat	HWAS-CG	*NRBP2*	14	961,099	968,482	0.00036	Blood
Protein	HWAS-H	*TFAP2C*	13	59,366,399	59,375,952	0.022	Embryo
Lactose	M-GWAS	*GPAA1*	14	750,608	753,850	7.7 × 10^−7^	Muscle
FCM	M-GWAS	*RASSF3*	5	49,114,394	49,190,776	0.0073	Macrophage
CH4 g/d	GWAS	*EXTL1*	2	127,052,889	127,070,881	0.025	Hypothalamus
CH4 DMI	GWAS	*APOBEC3H*	5	110,559,226	110,574,354	0.00178	Monocyte
CH4 ECM	HWAS-CG	*C9H6orf163*	9	62,541,669	62,569,565	0.0027	Oviduct
Acetate	HWAS-CG	*SPOCD1*	2	121,894,026	121,924,935	0.022	Uterus
Propionate	HWAS-CG	*MED29*	18	49,092,003	49,100,532	0.00021	Macrophage
Wolin	HWAS-CG	*MED29*	18	49,092,003	49,100,532	0.00036	Macrophage
DG	HWAS-CG	*IL33*	1	215,899,435	215,941,840	1 × 10^−6^	Colon
FC	HWAS-CG	*APCDD1*	6	97,992,228	98,026,768	0.0009	Hypothalamus
FI	HWAS-H	*SMPD1*	9	3,324,091	3,328,373	0.016	Hypothalamus

Milk yield (milk, fat, protein, lactose, and FCM), methane emissions (CH4 g/d, CH4 DMI, and CH4 ECM), rumen physiology (acetate, propionate, and wolin), and gut microbial composition (DG, FC, and FI).

**Table 4 ijms-25-06234-t004:** The top significant genes associated with complex traits according to SMR analysis.

Trait	Model	Gene	Chr	P_GWAS_	P_e-QTL_	P_SMR_	FDR	Tissue	P_HEIDI_
Milk	M-GWAS	*DGAT1*	14	7.42	15.29	5.28	0.003	Mammary	0.0005
Fat	HWAS-CG	*DGAT1*	14	4.25	16.60	3.56	0.626	Blood	6 × 10^−5^
Protein	HWAS-H	*KLF15*	22	4.54	6.23	2.87	0.283	Monocytes	0.8467
Lactose	M-GWAS	*DGAT1*	14	8.58	18.01	6.10	0.001	Liver	1.8 × 10^−3^
FCM	M-GWAS	*PLTP*	13	3.95	6.44	2.68	0.958	Muscle	0.00698
CH4 g/d	GWAS	*AOX1*	2	4.16	15.04	3.44	0.141	Adipose	-
CH4 DMI	GWAS	*DDX17*	5	4.73	5.74	2.84	0.927	Mammary	-
CH4 ECM	HWAS-CG	*PTBP3*	8	4.75	6.85	3.05	0.891	Macrophage	1.6 × 10^−5^
Acetate	HWAS-CG	*ZBTB8B*	2	3.96	15.21	3.32	0.381	Uterus	0.9203
Propionate	HWAS-CG	*SLC20A2*	27	3.09	28.16	2.87	0.929	Blood	0.1584
Wolin	HWAS-CG	*SLC20A2*	27	3.18	28.16	2.95	0.948	Blood	0.1820
DG	HWAS-CG	*FADS1*	23	4.55	25.60	4.01	0.52	Muscle	3 × 10^−6^
FC	HWAS-CG	*ENSSSCG00000037808*	4	4.39	13.09	3.49	0.271	Frontal cortex	8 × 10^−6^
FI	HWAS-H	*QRSL1*	1	2.75	8.56	2.24	0.737	Hypothalamus	7.3 × 10^−5^

The *p*-values of P_GWAS_, P_e-QTL_, and P_SMR_ are at the −log10 scale.

## Data Availability

All data generated or analyzed during this study are available from the corresponding author upon reasonable request.

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
