# Peer review of "Exploring the Interplay between the Hologenome and Complex Traits in Bovine and Porcine Animals Using Genome-Wide Association Analysis"

_ijms, 2024, doi:10.3390/ijms25116234_

Round 1

Reviewer 1 Report

Comments and Suggestions for Authors

IJMS-2941652- Peer-Review-Report-v1
In this paper, the authors explored the importance of integrating host and microbial genetics
to dissect complex traits of milk yield, methane emissions, rumen physiology, and gut
microbial composition using four different statistical models. The study unravelled the
hologenomics variability of important livestock traits, which will improve the understanding
of host-microbiome relationships and advance animal research. Specifically, the impact of
livestock on the environment, as the traits studied contribute to global warming and climate
change.
Abstract
The abstract provides an overview of the study's objectives, methods, results, and conclusions.
However, there are several areas for improvement:
a) The abstract contains repetitive content, with some sentences being duplicated. It is
essential to restructure the abstract and enhance readability.
b) The paper has acronyms, which may be unclear for readers unacquainted with the
field. To enhance ease of access for a considerable audience, it would be mandatory to
define these terms upon their first mention.
Introduction
a) The introduction should be better reorganised and structured so that the material
presented flows sequentially with each concept presented connecting logically with
the previous one.
b) The research gaps or the study's specific objectives are not clearly articulated. Clearly
stating the research gap, research questions, or objectives would provide readers with a
clear understanding of the study's purpose.
c) Some terminologies and acronyms, such as GWAS, LMM, OUT, TWAS, and
hologenomics, are presented without detailed explanations. The authors must provide
clear definitions or descriptions of these terms, mainly for readers unfamiliar with them.
Materials and Methods
a) The authors need to expand the justification for choosing specific datasets (e.g.,
Holstein cows and German Piétrain cows). Why were the above datasets selected over
others, and how representative are they of the larger population?
b) The authors must explain the rationale for choosing Bonferroni correction as the
threshold method in the statistical analysis. Why was it deemed more appropriate for
this study than other methods like FDR control?
c) The authors should ensure that all methods and software mentioned in the text are
appropriately cited for readers to refer to for further information.
Results:
a. All the acronyms of the genes in this section and throughout the text should be
italicised.
b. Lines 351-375: These two paragraphs look more or part of the discussion section. The
authors should consider deleting the two paragraphs from the result section and
incorporating them in the paper's discussion section.
c. Lines 421-424: The title of Table 3 is too long. The traits in parenthesis should deleted
from the title but could be described elsewhere.
d. Lines 424-425: The statement ‘The p-values were transformed as -log10(p-value)’ could
be placed under Table 3.
Discussion:
a. Some elements of the discussion are overly verbose and repetitive, which detracts from
the clarity. Streamlining the discussion and focusing on key points would improve
your readers' readability and engagement.
b. Line 469: Change ‘has’ to ‘have’.
c. Lines 469-471: The authors should ensure a consistent citation style throughout the text.
For instance, Cochran et al. has reported this SNP and CPSF1 gene's significant
association with milk production traits such as milk and fat yield, and fat and protein
percentage in Holstein cattle [46]. It should rather be ‘Cochran et al. [46] have reported
this SNP and CPSF1 gene's significant association with milk production traits such as
milk and fat yield, and fat and protein percentage in Holstein cattle’. This should be
checked throughout the text.
d. Lines 518-519: Aspergillus fumigatus should be italicised (Aspergillus fumigatus).
e. Lines 527: Sus scrofa should be italicised (Sus scrofa).
f. Lines 542: Bacteroides dorei should be italicised (Bacteroides dorei).
g. Line 550: Change ‘suggests’ to ‘suggested’.
Conclusion:
a. The conclusion appears too long and verbose; it should be summarised.
b. The conclusion offers a broad synopsis of the study's methods and findings without
delving into specific results or insights gained from analysing the complex traits. This
makes it difficult for readers to identify the key findings and their significance.
c. The study does not discuss limitations encountered, such as sample size, population
structure, technical concerns, or methodological constraints, that could impact the
reliability and validity of the findings.
d. Ponder on discussing the practical consequences of the study findings for livestock
breeding strategies, methane emission mitigation strategies, and advancing animal
health and productivity.
References: The authors did not follow the MDPI guidelines on referencing. They should
follow the guidelines when writing out the references used in the text.

Comments on the Quality of English Language

Minor and have been pointed out in the main report attached.

Reviewer 2 Report

Comments and Suggestions for Authors

Overview

The current manuscript entitled “Exploring the hologenome interplay with complex traits in livestock using genome-wide association analysis” explores the genetic and microbial factors of complex traits in Holstein cows and Piétrain pigs.

Using genome-wide association studies (GWAS), microbial GWAS (M-GWAS), and hologenome-wide association studies (HWAS), including CORE-GREML (HWAS-CG) and Hadamard product (HWAS-H) models, the study identifies significant SNPs influencing traits like milk yield, methane emissions, rumen physiology, and gut microbial composition. It highlights the intricate interactions between host genomes and microbiomes, providing insights into the genetic architecture underlying these agriculturally important traits.

Overall, this is a well-written manuscript. The objectives and rationale are clearly stated, and the statistical analyses are well performed and reported. The current figures and tables are sufficient and informative. The interpretation of the results and conclusions are supported by the data. The manuscript has emphasized the strengths and stated limitations. The manuscript structure and flow are satisfactory.

General comments and questions

The species under study were not mentioned until the Materials and Methods section (even the traits been investigated were mentioned before the species), this may hinder readability.

In Table 1, it is not clear whether these number are the sum of all the significant SNPs cross all 14 traits or not. In addition, I do not see a huge necessity to include this information where different multiple correction technics are considered as a metric to measure the data. Please justify this.

While the manuscript titled “livestock”, the study mainly used two livestock species/breeds, Holstein cows and Piétrain pigs. There seems to be a bit generalization issue, please justify this.

There is limited information regarding the how the traits are sampled/collected.

There is limited information reported to support replicability and/or reproducibility.

There is not experimental validation of the computational predictions reported in the study.

Formatting issues:

1.     there are extra or bigger spaces in multiple lines.

2.     Sometimes the font type and/or size are inconsistent.

Specific comments and questions

Line 41-43: "more than one factor" and "many genes and variants" both convey the idea of multiplicity, this is a bit redundant.

Line 88: the period may need to be placed outside the quotes.

Line 65: formatting issue. The spaces between words are tiny.

Line 125, 135, 291, 500, 501: extra space in the sentence.

Line 345: the definition of genomic range can be further clarified, for example, 100kb upstream and 100kb downstream, respectively? And how to assign candidate genes into the region, fully located in the region or just overlapping sections would be allowed?

Round 2

Reviewer 1 Report

Comments and Suggestions for Authors

IJMS-2941652-peer-review-report-v2

The authors have made significant progress in implementing the suggestions earlier presented to them. With a few more adjustments, this paper has the potential to move to the next stage of the review and be accepted for publication. My further suggestions are listed below:

a.     Line 96: Write the word ‘animal’ in its plural form (animals).

b.     Line 184: For correctness, the word ‘effect’ should be ‘effects’.

c.      Line 186 Add the article ‘a’ before the word ‘polygenic’.

d.    Lines 349-350: For clarity, this sentence should be paraphrased. Suggestion: Four models were used to analyse methane production (CH4 g/d), yield (CH4 DMI), and intensity (CH4 ECM) in cattle emissions (Figure 4). 

e.     Lines 514-517: This sentence includes an incomplete comparison, consider rewriting to complete the comparison.

f.      Line 558: Italicise these two genes DGAT1 and SLC39A4.

g.     All the abbreviations of the different genes listed in Tables 2-4 should be italicised.

h.     The authors have failed to follow the MDPI referencing format. For example, Reference number 1 is written as follows: Raymond, B., et al., Using prior information from humans to prioritize genes and gene-associated variants for complex traits in livestock. PLoS Genetics, 2020. 16(9): p. e1008780. All the authors' names must be written out except if they are more than 4 or 5; in that case, subsequent names may not appear. The name of the journal should be written in italics. So reference number 1 should be: Raymond, B.; Yengo, L.; Costilla, R.; Schrooten, C....Visscher P.M. Using prior information from humans to prioritize genes and gene-associated variants for complex traits in livestock. PLoS Genet. 2020, 16(9):e1008780. doi: 10.1371/journal.pgen.1008780. PMID: 32925905; PMCID: PMC7514049. It is very important that this is not overlooked.

Comments on the Quality of English Language

This is already pointed out in the attached report.
